# *The SmartSleep Experiment*: Evaluation of changes in night-time smartphone behavior following a mass media citizen science campaign

**Thea Otte Andersen** ⊙*, **Agnete Skovlund Dissing, Tibor V. Varga** ⊙**, Naja Hulvej Rod**

Section of Epidemiology, Department of Public Health, University of Copenhagen, Copenhagen, Denmark

* than@sund.ku.dk

## Abstract

The increasing 24-hour smartphone use is of public health concern. This study aims to evaluate whether a massive public focus on sleep and smartphone use generated through a large-scale citizen science project, the *SmartSleep Experiment*, influence participants' night-time smartphone behavior. A total of 8,894 Danish adults aged 16 and above participated in the *SmartSleep Experiment*, a web-based survey on smartphones and sleep behavior. The survey was carried out for one week in 2018, combined with an extensive national mass media campaign focusing on smartphone behaviors and sleep. A follow-up survey aimed at evaluating whether survey-participants had changed their night-time smartphone behavior was carried out two weeks after the campaign. A total of 15% of the participants who used their smartphone during sleep hours at baseline had changed their night-time smartphone behavior, and 83% of those indicated that they used their smartphone less at follow-up. The participants who had changed their smartphone behavior had primarily taken active precautions to avoid night-time smartphone use, e.g., activating silent mode (36%) or reduced their smartphone use before (50%) and during sleep hours (52%). The reduction in sleep problems (54%), recognition of poor smartphone behavior (48%), and the increased focus on night-time smartphone use (42%) were motivational factors for these behavior changes. Using citizen science and mass media appeared to be associated with changes in night-time smartphone behavior. Public health projects may benefit from combining citizen science with other interventional approaches.

## Introduction

Smartphones have become an integrated part of everyday life, and the increasing and widespread use of smartphones is an inevitable trend in today's digitized society. Smartphones are frequently used around the clock [1–3]. Previous studies have shown that night-time smartphone use is related to poor sleep quality and shorter sleep duration [1, 3–6]. Furthermore, a recent randomized trial with 38 college students found that restricting mobile phone use

**Data Availability Statement:** The data set contains personally identifiable and sensitive survey data information. We are therefore not allowed to make them publicly available according to the Danish

Protection Agency (Danish data protection legislation (datatilsynet.dk)) and Danish law. Inquiries for secure data access under conditions stipulated by the Danish Data Protection Agency should be directed at the data manager of the SmartSleep project (liks@sund.ku.dk) or principle investigator of the SmartSleep project Professor Naja Hulvej Rod (nahuro@sund.ku.dk).

**Funding:** The project was funded by the Independent Research Fund Denmark (grant number 7025-00005B). The funder had no role in study design, data collection and analysis, decision to publish, or preparation of the manuscript.

**Competing interests:** The authors have declared that no competing interests exist.

before bedtime reduced sleep latency and increased sleep duration [7]. Because poor and short sleep is a well-known risk factor for obesity, cardio-vascular diseases, diabetes, and mortality [8, 9], the increasing 24-hour smartphone use highlights a pressing public health issue. Thus, there is a need for public health prevention strategies to change night-time smartphone behavior to eventually improve sleep behavior and health.

Citizen science, defined as public participation in scientific research in which the public are engaged directly in one or more of the research processes [10, 11] has proven to be an effective strategy to maximize the social impacts of research [12]. It has also been highlighted as a strategy to improve the linkage between research, education, and action [13, 14]. The use of citizen science has expanded in recent years, especially in biomedical and public health research [15, 16]. Citizen science in public health research helps generate scientific knowledge and new insights into complex problems [12, 17]. Also, it makes it possible to investigate the nuanced understandings of public health trends and mechanisms [13, 14]. Furthermore, it has been argued that citizen science and mass media campaigns may serve as a method of health promotion, as participants might increase their health literacy, change their attitudes and norms, and increase their awareness of their own health behaviors [16–19]. All these changes may lead to subsequent behavior changes [17, 20–22].

Only a few public health citizen science projects to date have evaluated whether the citizen science approach *per se* has an interventional effect on the participants' health behavior [18, 23]. This may be due to the fact that there are no commonly established measures for evaluating citizen science projects or mass media campaigns and collecting evidence of their impact [20]. Therefore, there is only sporadic evidence in the literature on potential behavioral interventional impacts of mass media campaigns and participating in citizen science public health projects.

In 2018, the *SmartSleep Experiment*, a web-based survey about smartphone use and sleep, was carried out in collaboration with the Danish Broadcasting Corporation (DR) using a citizen science approach. During one week of data collection, DR created a national mass media campaign focusing on night-time smartphone behaviors and sleep, which was used as a vehicle to create attention and public interaction around the research project. The public attention helped form a framework for recruiting more than 25,000 participants for an online survey. This paper aims to evaluate whether the massive public focus on night-time smartphone use and sleep using mass media during the *SmartSleep Experiment* was associated with reductions in night-time smartphone use among participants. We will specifically 1) explore whether participants had changed their night-time smartphone behavior and in which direction, and 2) examine behavioral and motivational factors that may have influenced their smartphone behavior changes among a subsample of 8,894 people who participated in the *SmartSleep Experiment* and were followed-up immediately after the massive public focus on smartphone use and sleep.

## Material and methods

### The SmartSleep Experiment

The *SmartSleep Experiment* was carried out for one week in November 2018. The core scientific element of the *SmartSleep Experiment* was an online survey aimed at documenting smartphone use during sleep hours in the adult population. The participants were actively involved in the data collection by filling out a survey, which was an integrated part of the media campaign. This was combined with direct individual feedback to the participants and real-time presentation of preliminary results from the data collection to all radio listeners during the week of the campaign. The participants accessed the survey either via the *SmartSleep* website (www.smartsleep.ku.dk) or a link on DR's website (www.dr.dk/soverdugodt) to which they

were directed from the radio programs, teasers, online articles, and social media. The participants had to be 16 years or older and understand written Danish. The online survey contained information on demographics, smartphone use, sleep behavior, and health-related questions.

The present study was approved by the Danish Data Protection Agency through the joint notification of The Faculty of Health and Medical Sciences at The University of Copenhagen (record no 514-0237/18-3000 and 514-0288/19-3000). All data was handled according to GDPR guidelines. Survey-based studies do not require ethical approval by the Danish National Committee on Health Research Ethics according to Danish Law. Written informed consent was obtained from all participants. The participants were informated about the purposes of the research and their rights to withdraw.

## Direct personal feedback

To motivate people to participate in the survey, they received immediate personal feedback on their night-time smartphone use after completing the survey. The personal feedback included a description of the participant's night-time smartphone use and a comparison with results from the Danish Regional Health Survey 2017 [24]. The comparison included the proportion of participants who did not get enough sleep to feel rested and the proportion of participants who stated that the reason for insufficient sleep was the use of or disturbance of their smartphone according to various age groups (16–24, 25–34, 35–44, 45–54, 55–64, 65–74, 75+). Please see an example of a personal feedback in S1 Fig.

## Massive public media coverage on night-time smartphone use and sleep

During the *SmartSleep Experiment* week, DR created a public campaign, "Do you sleep well?" on several of their platforms including radio programs, their website, and social media pages. The three-hour national morning radio show "Good morning P3" on the radio channel DR P3 was the main driving force of the *SmartSleep Experiment*. Each morning during the week, the radio program focused on smartphone use or sleep from different perspectives and encouraged listeners to engage in discussions and participate in the survey. To further create a feedback process between the researchers and participants, preliminary results–"The results of the day" were generated by the researchers at the University of Copenhagen each day and reported live on the radio. Furthermore, the researchers were live in the studio every morning during that week. The radio channel DR P3 has primarily young to middle-aged listeners and was the second most listened to radio channel in Denmark in 2018. Media analyses from DR show that during the *SmartSleep Experiment* week, 1.5 million people listened on P3. Furthermore, each morning during the week, approximately 489,000 listened to the morning radio program "Good Morning P3" and there were 807,000 unique listeners (15% of the Danish population aged 12 and above) for the entire week to "Good Morning P3"

Local radio programs at the regional radio channel DR P4 were also used as a platform to engage participants to improve geographical representation and generate further attention to the *SmartSleep Experiment*. The researchers were also interviewed about the project on the local radio programs on DR P4. The radio programs were also focused on various themes related to smartphone use and sleep. The regional radio channel P4 has primarily middle-aged listeners, and it is the most listened to radio channel covering ten regions of Denmark. During the week of the *SmartSleep Experiment*, approximately 2.7 million individuals (53% of the Danish population aged 12 and above) listened to the P4 regional radio channels.

More than 20 news and feature articles focusing on the importance of sleep for health and well-being and on how night-time night-time smartphone use may result in disturbed sleep were published during the week of the experiment on DR's website and their social media

pages, including Twitter, Facebook, and Instagram. Furthermore, an animated video was published on DR's online platforms demonstrating the potential health problems associated with night-time smartphone use. The video also encouraged people to participate in the survey.

A countdown timer was set up on DR's website on the last day of the *SmartSleep Experiment* showing the time left for respondents to answer the survey. Furthermore, various radio programs at DR encouraged their listeners to participate in the survey before the end of the experiment at midnight.

### Follow-up survey

Two weeks after the *SmartSleep Experiment*, participants who previously indicated they wanted to participate in future studies received an online follow-up survey via e-mail. The follow-up survey aimed to evaluate whether the participants had changed their smartphone behavior after the massive public focus on smartphone use and sleep. Specifically, the follow-up survey focused on night-time smartphone use, changes in night-time smartphone use, and motivational factors for smartphone behavior changes. Up to three reminders were sent to those not responding to the first e-mail.

### Study population and data collection

A total of 25,135 Danish adults participated in the core survey on the *SmartSleep Experiment* during the citizen science week. Two weeks after the *SmartSleep Experiment*, 12,348 (49%) participants who had indicated that they wanted to participate in future studies received the online follow-up survey. In total, 8,911 responded to the follow-up survey (72% response rate). Seventeen participants were excluded from the study population because they did not have a smartphone. Thus, 8,894 participants were eligible for the analyses. S2 Fig shows the flowchart of the study population. S1 Table shows characteristics of individuals who at baseline did not agree to participate in future studies (n = 12,887), individuals who agreed at baseline but did not participate in the follow-up study (n = 3,337), and individuals who participated in the follow-up study (n = 8,911). Those who participated in the follow-up study were generally older, had higher educational level, and were slightly less likely to use their smartphone during sleep hours at baseline than those who did not participate.

### Measurements

*Perceived changes in night-time smartphone use* were assessed by the question: "Have you changed your smartphone behavior at night after you have answered the last *SmartSleep* survey?"

a. Yes, I have changed them

b. I have tried changing them, but it was unsuccessful

c. No, I have not changed behavior.

*The direction of change* was assessed by asking the participants whether they check their smartphone during sleep hours when they would typically sleep more or less since completion of the *SmartSleep* survey

a. More often

b. Less often

c. at the same level as before

**Behavioral factors.** Those participants who reported to have changed their night-time smartphone behavior were asked how they had changed their behavior; response options: a) I set my smartphone on silent mode, flight mode or do not disturb, b) I turn off my smartphone, c) I place my smartphone out of reach, d) I reduce my smartphone use before falling asleep, e) I reduce my smartphone use during sleep hours, f) I use an analog alarm clock instead of the alarm clock on my smartphone, and g) other. The participants could select multiple response options.

**Motivational factors.** Those participants who changed their night-time smartphone behavior were asked about motivational factors, response options: a) Increased focus on night-time smartphone use in the media, e.g., the theme "Do you sleep well?" on DR, b) I have received new knowledge about the health consequences of poor sleep, c) I have discussed with people close to me about my smartphone behavior, d) I felt that my smartphone behavior was bad for me, e) I wanted to reduce my sleep problems, and f) other. The participants could select multiple response options.

*Age (10-year bands)*, *gender*, *educational level* (low (primary school); medium (upper secondary school; technical vocational education); high education (short, medium, and long cycle higher education); and other based on the International Standard Classification of Education 2011 [25]), *occupational status* (employed; student; unemployed; outside labor market; long-term sick leave; other), *cohabitation* (living alone versus not living alone), *baseline night-time smartphone user (yes versus no)* and *sleep quality* was obtained in the baseline survey. *Baseline night-time smartphone user* was assessed by asking how often the smartphone was used after falling asleep and during sleep hours within the past three months with the following response options: every night or almost every night; several nights a week; several nights a month or less; and never. Smartphone use was defined in the baseline survey and referred to both short and long activation of the smartphone. *Sleep quality* was assessed using a Danish translation of a validated short version of the Karolinska Sleep Questionnaire (KSQ) [26]. The KSQ includes four items covering the frequency of poor sleep quality rated from never to every night or almost every night. KSQ ranges from one to five and a higher score indicates poorer sleep quality.

## Analytical strategy

First, we report the distribution of participation during the week of the *SmartSleep Experiment* among 25,135 participants. We plotted the distribution of participants according to the date and time each participant finished the online survey. Of these, 8,894 participants also took part in the follow-up survey. We described characteristics of the participants according to their changes in night-time smartphone behavior to explore differences between the groups. Differences in categorical variables were tested with a chi-squared test and for differences in continuous variables, we used ANOVA.

To investigate whether participants who used their smartphone during sleep hours at baseline had changed their night-time smartphone behavior after the *SmartSleep Experiment*, we restricted the population to those who reported using their smartphone during sleep hours at baseline (n = 4,926). We calculated the proportions who reported to have changed their smartphone behavior at follow-up. Finally, we assessed the proportions of behavioral and motivational factors for changes in night-time smartphone behavior among those who reported to had changed their night-time smartphone behavior to explore the mechanisms behind behavior changes. All analyses were performed using R version 3.6.3.

## Results

### Distribution of participation

A total of 25,135 people participated in the *SmartSleep Experiment*. Fig 1 shows the distribution of participation including key events during the week of the *SmartSleep Experiment*. The

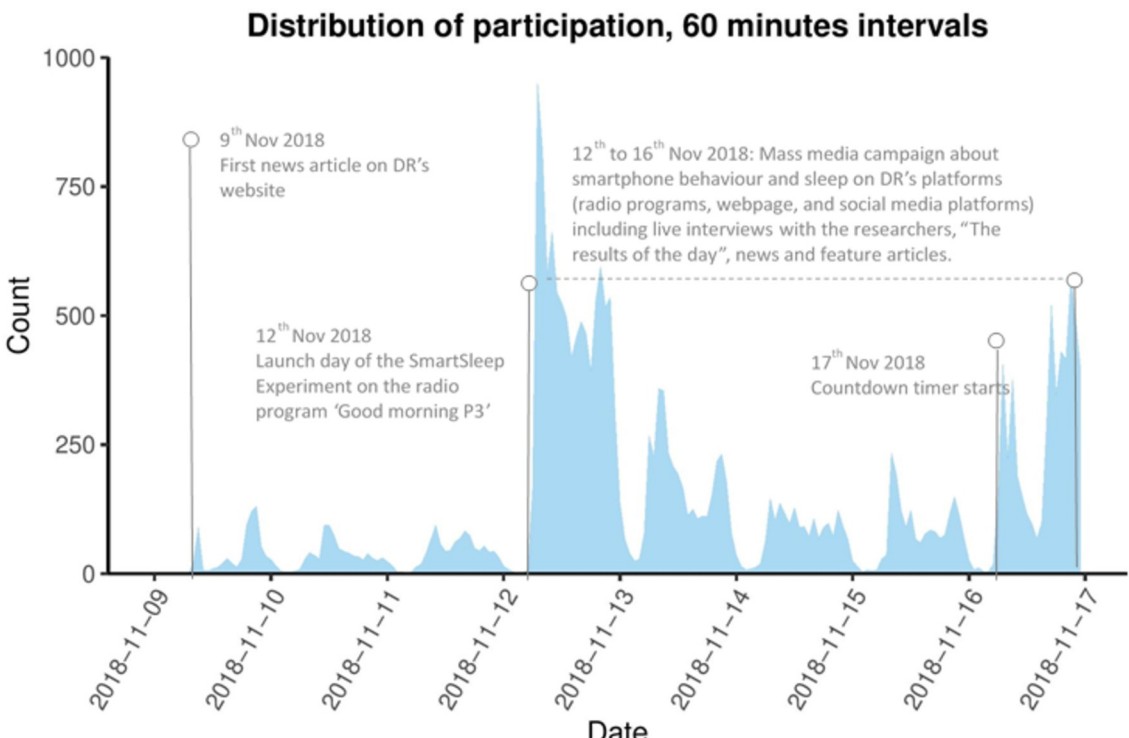

**Fig 1. Distribution of participation during the week of the *SmartSleep Experiment* among 25,135 participants.**

daily distribution of participants clearly shows an impact of the partnership with DR. While the survey was online on DR's website and on the website of the *SmartSleep Experiment* from day 1 (9 Nov 2018), only a few articles were published on DR between day 1 and 3 (9–11 Nov 2018). Thus, only 2,448 responded to the survey before 12 Nov 2018. The fourth day (12 Nov 2018) marked the launch day of the *SmartSleep Experiment* in various radio programs including "Good Morning P3!" There is a corresponding major increase in the survey participants on this day (total 9,478 participants on 12 Nov 2018). Between 12–16 Nov 2018, participation was primarily observed in the morning and the afternoon/evening. Notably, 10% of all participants filled in the survey at night between 11 PM and 7 AM. There was a peak in participation at the end of the experiment on the final day of the survey (16 Nov 2018), probably because of the countdown presented on all media platforms on this day.

## Characteristics of participants with changed smartphone behavior

Table 1 shows the characteristics of the study population according to whether they have changed their smartphone behavior at follow-up. Almost one in ten of the participants (9%) indicated that they had changed their night-time smartphone behavior after participating in the *SmartSleep Experiment*. Younger participants were more likely to change or try to change their smartphone behavior than older participants. Slightly more men than women did not change smartphone behavior. More participants with high education did not change their smartphone behavior compared to participants with low or medium education. Additionally, more participants with low or medium education tried to change their smartphone behavior, but were unsuccessful compared to participants with high education. Participants who were employed or outside labor market were more likely not to change their smartphone behavior, while unemployed and students were more likely to change or try to change their smartphone

**Table 1. Characteristics of the study population according to changed night-time smartphone behavior among 8,894 participants in the *SmartSleep Experiment*.**

| | Total n = 8,894 | Changed behavior n = 838 (9%) | Tried to change behavior, but was unsuccessful n = 549 (6%) | Have not changed behavior n = 7,507 (85%) | P-value |
|---|---|---|---|---|---|
| **Age**, n (%) | | | | | |
| 16–25 | 923(10) | 102 (11) | 119 (13) | 702 (76) | |
| 26–35 | 1,620 (18) | 177 (11) | 130 (8) | 1,313 (81) | |
| 36–45 | 1,710 (19) | 152 (9) | 112 (7) | 1,446 (85) | |
| 46–55 | 2,109 (24) | 202 (10) | 116 (6) | 1,791 (85) | |
| 56–65 | 1,702 (19) | 139 (8) | 63 (4) | 1,500 (88) | |
| +65 | 830 (9) | 66 (8) | 9 (1) | 755 (91) | <0.001 |
| **Gender**[a], n (%) | | | | | |
| Female | 5,387 (61) | 547 (10) | 376 (7) | 4,464 (83) | |
| Male | 3,496 (39) | 289 (8) | 173 (5) | 3,024 (87) | <0.001 |
| **Educational level**[b], n (%) | | | | | |
| Low | 380 (4) | 38 (10) | 40 (11) | 302 (80) | |
| Medium | 2,042 (23) | 205 (10) | 157 (8) | 1,680 (82) | |
| High | 6,283 (71) | 573 (9) | 340 (5) | 5,370 (86) | |
| Other | 189 (2) | 22 (12) | 12 (6) | 155 (82) | |
| **Occupational status,** n (%) | | | | | |
| Employed | 5,853 (66) | 530 (9) | 319 (6) | 5,004 (86) | |
| Student | 1,218 (14) | 135 (11) | 142 (12) | 941 (77) | |
| Unemployed | 233 (3) | 30 (13) | 24 (10) | 179 (77) | |
| Outside labor market | 1,144 (13) | 96 (8) | 25 (2) | 1,023 (89) | |
| Long-term sick leave | 119 (1) | 10 (8) | 16 (13) | 93 (78) | |
| Other | 327 (4) | 37 (11) | 23 (7) | 267 (82) | <0.001 |
| **Baseline night-time smartphone user,** n (%) | | | | | |
| Yes[c] | 4,926 (55) | 720 (15) | 509 (10) | 3,697 (75) | |
| No[d] | 3,968 (45) | 118 (3) | 40 (1) | 3,810 (96) | <0.001 |
| **Cohabitation**, n (%) | | | | | |
| Living alone | 1,935 (22) | 195 (10) | 141 (7) | 1,599 (83) | |
| Not living alone | 6,959 (78) | 643 (9) | 408 (6) | 5,908 (85) | 0.029 |
| **Sleep quality**, mean (SD) | 2.8 (1) | 3.1 (0.9) | 3.3 (0.9) | 2.7 (1) | <0.001 |

[a] NA, n = 11

[c] Low education: Primary school; Medium education: Upper secondary school or technical vocational education; High education: Short, medium, or high cycle higher education

[c] Reporting 'every night or almost every night', 'several nights a week' or 'several nights a month or less'

[d] Reporting 'never'.

behavior. As expected, participants who used their smartphone during sleep hours at baseline were more likely to change or try to change their smartphone behavior than those who did not use their smartphone during sleep hours. S2 Table shows the socio-demographic differences in baseline night-time smartphone use. It appears that participants with high education, employed people, and participants outside labor market were less likely to use their smartphones during sleep hours at baseline, while students were more likely to use their smartphones during sleep hours at baseline. Participants who changed or tried to change their

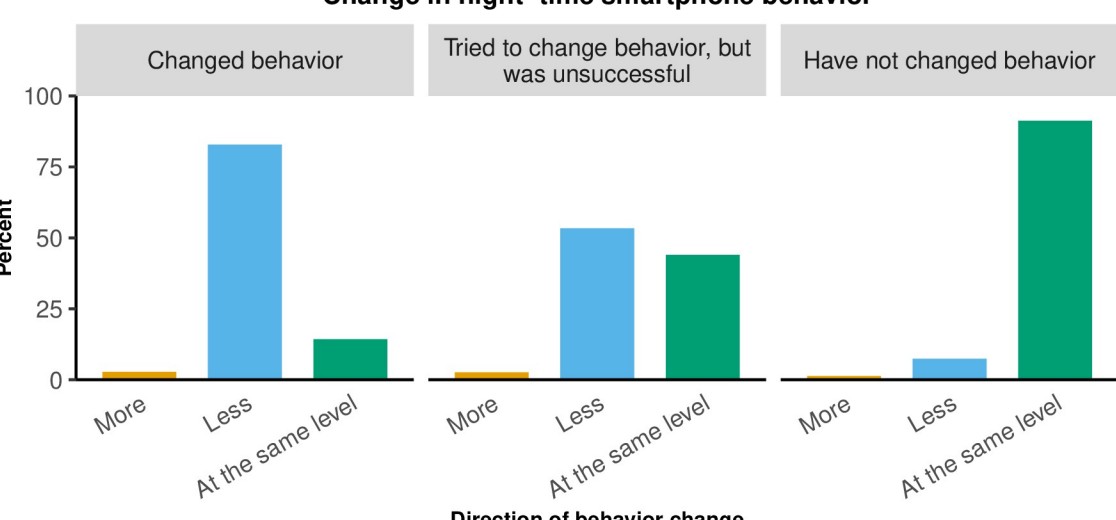

**Fig 2. Change in night-time smartphone behavior according to the direction of change among 4,926 participants who used their smartphone during sleep hours at baseline.**

smartphone behavior had poorer sleep quality compared to participants who did not change their smartphone behavior.

## Change in night-time smartphone use and the direction of change at follow-up among baseline night-time smartphone users

In total, 720 participants (15%) out of 4,926 participants who used their smartphone during sleep hours at baseline reported having changed their night-time smartphone behavior after the *SmartSleep Experiment* (Table 1). As demonstrated in Fig 2, 83% of these participants stated that they used their smartphone less during sleep hours. Also, 91% of those who indicated did not change behavior at follow-up stated that they used their smartphone at the same level as before the *SmartSleep Experiment*.

## Behavioral and motivational factors for changes in night-time smartphone behavior

Approximately half of the participants who changed their night-time smartphone behavior reduced their smartphone use before falling asleep at night (50%) and/or during sleep hours (52%), as shown in Fig 3. Furthermore, around one-third of the participants had taken specific preventative precautions to avoid night-time smartphone use. E.g. 36% indicated that they had set their smartphone on silent mode, flight mode, or 'do not disturb' during sleep hours, and 29% reported that they had placed their smartphone out of reach during sleep hours.

Of the 720 participants who had changed their night-time smartphone behavior, more than half (54%) changed night-time smartphone behavior because they wanted to reduce their sleep problems (Fig 4). In total, 48% changed behavior because they felt that their smartphone behavior was unhealthy, and 42% changed behavior due to the increased public focus on smartphone and sleep behavior. Furthermore, 28% changed their night-time smartphone behavior because they had received more knowledge about the health consequences of poor sleep.

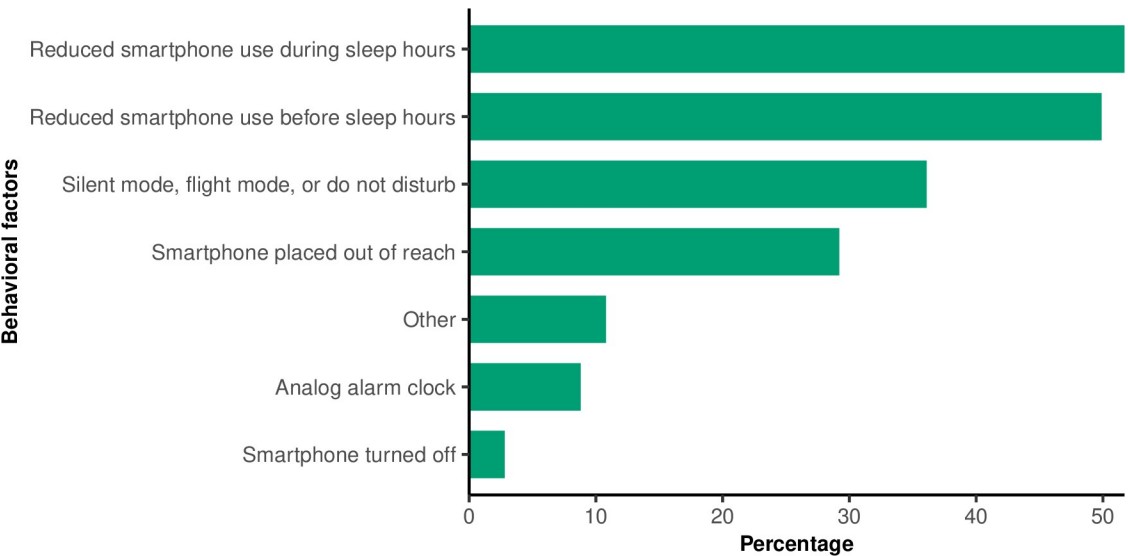

**Fig 3. The behavioral factors for changes in night-time smartphone behavior among 720 participants who changed behavior in the *SmartSleep Experiment*.**

## Discussion

In a large-scale citizen science project, we show that partnering with a public media platform greatly influenced the participation in the *SmartSleep Experiment*. More than 25,000 Danish adults participated within one week, and the pattern of participation corresponds with the timing of media exposure. Furthermore, we find that this mass media campaign and the citizen science approach with direct interaction between scientists and radio listeners during the *SmartSleep Experiment* appeared to be associated with changes in the participants' night-time smartphone behavior. In the current study, 15% of the participants who used their smartphone

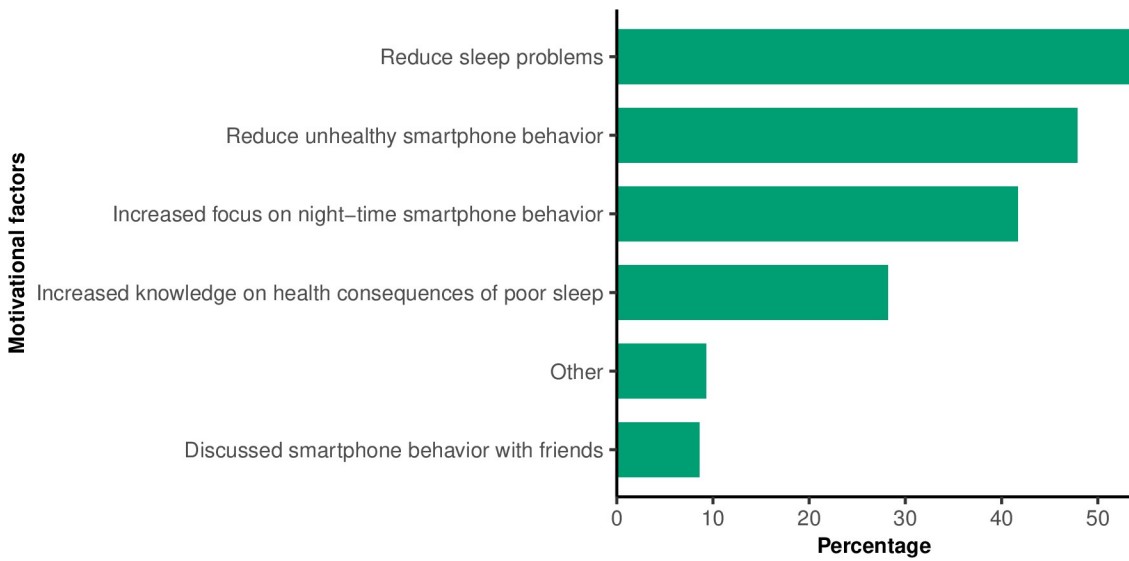

**Fig 4. The motivational factors for changes in night-time smartphone behavior among 720 participants who changed behavior in the *SmartSleep Experiment*.**

during sleep hour at baseline indicated that they had changed their night-time smartphone behavior, and 83% of these indicated that they used their smartphone less during sleep hours, but the underlying mechanisms are still unsettled.

A Dutch study from 2017 evaluated the impacts of participation in two public health citizen science projects using qualitative and quantitative methods [18]. The authors suggest that the citizen science projects functioned as health promotion interventions as the citizen scientists reported changed lifestyle behavior based on the evaluations. An environmental study from 2013 explored the linkage between citizen science and the potential impacts on conservation attitudes and behavior [13]. Based on an evaluation of two citizen science projects concerning environmental conservation, the study found that participation in citizen science projects had influenced the participants' conservation behavior. To explain the perceived impacts of conservation attitudes and behavior, the authors proposed a theoretical model. They argue that participants in citizen science projects may raise awareness of their own behavior leading to changes in attitudes and behavior [13]. Based on the proposed model, behavioral intention and willingness to change behavior are key factors in successfully changing night-time smartphone behavior.

In the current study, more than half of the participants who changed their night-time smartphone behavior were motivated by wanting to reduce their sleep problems. Sleep problems are a prevalent and increasing public health issue in adult populations [27, 28] and evidence has shown that sleep problems are related to adverse short- and long-term health effects including poor mental health, risk-taking behavior, cardiovascular diseases, diabetes, and mortality [8, 9]. Furthermore, several studies have shown that night-time smartphone use is related to poor sleep [1, 5, 6]. The massive public focus on sleep and night-time smartphone use in the mass media during the *SmartSleep Experiment* may have increased awareness of sleep and smartphone behavior, which may be an effective strategy to improve sleep behavior [29]. Participants who changed their night-time smartphone behavior reported that they wanted to reduce unhealthy smartphone behavior. This result indicates that increased awareness and knowledge are key factors contributing to changes in the attitudes and behavior of participants.

Night-time smartphone behavior changes may also be influenced by the individual perception of behavior control and the perceived ease or difficulty of changing the behavior. Around half of the participants who changed night-time smartphone behavior stated that they have reduced their smartphone use before falling asleep at night and during sleep hours when they would typically sleep. Moreover, around one-third of the participants with changed smartphone behavior have taken specific preventative precautions to reduce night-time smartphone use, e.g., activating silent mode or placing the smartphone out of reach. This result indicates that relatively simple precautionary behaviors may influence night-time smartphone use and contribute to smartphone behavior changes. Thus, using citizen science and mass media campaign in terms of recruiting participants and creating attention around sleep and smartphone behavior may positively impact the participants' night-time smartphone behavior.

While the citizen science approach is promising, still 75% of those who used their smartphone during sleep hours at baseline indicated that they did not change their night-time smartphone behavior following the campaign. The massive public focus on sleep and smartphone use in the mass media was carried out for only one week, and this timeframe may be short to affect participants' night-time smartphone behavior. Moreover, it has previously been shown that citizen science projects that are 'scientist-led' and where the participants primarily serve as data collectors have a lower potential for behavior changes compared to projects where participants are involved in more or all aspects of the scientific processes [10, 11]. Furthermore, while short-term behavior changes may be achieved using mass media campaigns and citizen science, the long-term effects may be more difficult to maintain. It has previously

been proposed that longer and more intense mass media campaign are likely to be more effective to maintain long-term behavior changes [19].

## Strength and limitations

This study provides novel insights into the behavioral effects on night-time smartphone behavior after participating in a citizen science project and being exposed to a mass media campaign. Furthermore, the present study elucidates the behavioral and motivational factors that may lead to such a change in night-time smartphone behavior. The results of the present study are based on perceived changes in night-time smartphone behavior and thus, it would be interesting to explore the actual changes in night-time smartphone use including the size of the interventional effect on night-time smartphone use. However, this was not possible in the present study as the measures on night-time smartphone use were not directly comparable in the two surveys and the measure on night-time smartphone use does not allow for assessment of the size of the changes. Even though we cannot determine the direct interventional effects from participating in the *SmartSleep Experiment*, the present findings give important insights into the connection between citizen science, mass media, and interventional behavior impacts, which may be of value in future prevention strategies to improve sleep and smartphone behavior.

Using mass media in the recruitment of participants have great potential, as more than 25,000 adults participated in the study within one week. All participants in the *SmartSleep Experiment* were exposed to at least parts of the media campaign, as this was the platform from which they were recruited. Unfortunately, we do not have specific information on how many elements of the campaign each person were exposed to, or how much of the information they recalled after the two week follow-up period. Furthermore, this recruitment strategy may also create selection problems as the selection mechanisms into the study are not transparent, and the participants are likely not representative of the Danish adult population. Moreover, participants who agreed to participate in the follow-up study may be more likely to change their smartphone behavior compared to those who did not want to participate, which may have resulted in a slight overestimation of the effect of the intervention. Furthermore, the direct personal feedback after completing the survey varied across age groups, and this may have influenced age-specific findings in likelihood of changing night-time smartphone use. According to the S1 Table, the participants in the follow-up study were older and had a higher education than those who did not participate in the follow-up study. For future studies, it would be interesting to explore whether using mass media campaigns to promote behavior changes would be more/less effective in younger populations. Moreover, all study participants have been exposed to the extensive mass media campaign about smartphone and sleep behavior on DR. Thus, these limitations make it difficult to investigate the direct effects of using citizen science and mass media.

## Conclusion

This study shows that massive media attention and the citizen science approach with direct interaction between scientists and radio listeners is associated with subsequent changes in night-time smartphone behaviors. Increased knowledge and awareness seemed to be the key motivational drivers. Public health projects may benefit from exploring the citizen science approach in combination with other interventional approaches.

## Supporting information

**S1 Fig. An example of the direct personal feedback after answering the baseline questionnaire.**
(TIFF)

**S2 Fig. Flowchart of the study population.**
(TIFF)

**S1 Table. Characteristics of individuals who participated in the follow-up study and individuals who did not participate in the follow-up study.**
(PDF)

**S2 Table. Baseline night-time smartphone use and socio-demograhic factors.**
(PDF)

## Author Contributions

**Conceptualization:** Thea Otte Andersen, Agnete Skovlund Dissing, Tibor V. Varga, Naja Hulvej Rod.

**Data curation:** Agnete Skovlund Dissing.

**Formal analysis:** Thea Otte Andersen, Agnete Skovlund Dissing, Tibor V. Varga, Naja Hulvej Rod.

**Methodology:** Thea Otte Andersen, Agnete Skovlund Dissing, Tibor V. Varga, Naja Hulvej Rod.

**Project administration:** Agnete Skovlund Dissing, Naja Hulvej Rod.

**Supervision:** Naja Hulvej Rod.

**Visualization:** Thea Otte Andersen.

**Writing – original draft:** Thea Otte Andersen, Naja Hulvej Rod.

**Writing – review & editing:** Agnete Skovlund Dissing, Tibor V. Varga, Naja Hulvej Rod.

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
