## [Decision Letter · Decision Letter 0]

26 Feb 2021

PONE-D-21-02819

The SmartSleep Experiment: The interventional effects of using a citizen science approach and mass media to change smartphone and sleep behaviors

PLOS ONE

Dear Dr. Andersen,

Thank you for submitting your manuscript to PLOS ONE. After careful consideration, we feel that it has merit but does not fully meet PLOS ONE’s publication criteria as it currently stands. Therefore, we invite you to submit a revised version of the manuscript that addresses the points raised during the review process.

In the revised version of the paper, besides addressing the reviewers' comments listed at the bottom of this email, please provide more details related to the results you obtained, while keeping the scientific rigor related to data presentation. Please discuss the limitations of the study and the impact of the research on the society.

We look forward to receiving your revised manuscript.

Kind regards,

Camelia Delcea

Academic Editor

PLOS ONE

Journal Requirements:

3.We note that you have indicated that data from this study are available upon request. PLOS only allows data to be available upon request if there are legal or ethical restrictions on sharing data publicly. For information on unacceptable data access restrictions, please see http://journals.plos.org/plosone/s/data-availability#loc-unacceptable-data-access-restrictions.

4.We note that the grant information you provided in the ‘Funding Information’ and ‘Financial Disclosure’ sections do not match.

Reviewers' comments:

Reviewer's Responses to Questions

**Comments to the Author**

1. Is the manuscript technically sound, and do the data support the conclusions?

Reviewer #1: No

Reviewer #2: No

Reviewer #3: Yes

2. Has the statistical analysis been performed appropriately and rigorously? 

Reviewer #1: Yes

Reviewer #2: No

Reviewer #3: Yes

3. Have the authors made all data underlying the findings in their manuscript fully available?

Reviewer #1: Yes

Reviewer #2: No

Reviewer #3: Yes

4. Is the manuscript presented in an intelligible fashion and written in standard English?

Reviewer #1: Yes

Reviewer #2: Yes

Reviewer #3: Yes

5. Review Comments to the Author

Reviewer #1: This manuscript examines the impact of a massive public media campaign on smartphone use at night in adults. This study suggests that public health efforts can be made through the mass media that can influence behavior, at least in the short term. I provide several questions regarding the study methods and conclusions along with recommendations for improving the paper below.

Introduction

• In the introduction, there is good discussion of the use of citizen science to conduct research, however this section could benefit from a discussion of the impact of smartphone use on sleep behavior and past interventions addressing this public health problem.

• The study purports to assess intervention effects on both smartphone use at night and sleep behavior but none of the study measures assess sleep behavior (e.g., sleep duration, seep onset latency, etc.), making the second aim impossible.

Methods

• The method used for recruiting participants presents a significant limitation to drawing conclusions from the study findings. Because only those who indicated interest in participating in research following the citizen science intervention, a sample bias likely exits because people who agree to participate in the may have been more likely to have changed their behavior. It is difficult to conclude that the intervention produced the change given this sampling bias.

• Another significant limitation is the inability to determine the size of the effect of the intervention on smartphone use at night. The four response choices in the measure do not allow for assessment of the size of the intervention effect.

• The utility of the measure of smartphone use is unclear because there are no reported studies assessing the psychometrics of this measure. Relatedly, the paper could be improved by reporting reliability and validity for the current study sample.

• Regarding the question assessing “behavioral interventions”, were participants allowed to select more than one of the options (e.g., used an analog clock as alarm AND placed my phone out of reach) or were they asked to select only one option? It could be helpful to clarify this with the “motivational factors” as well.

• Is there any additional information the authors could give about the rationale for including age, gender, and education level in the survey while they do not report other possible variables that could impact sleep (e.g., sleep disorders; socioeconomic status)?

Discussion

• It is interesting how many people in the study sample endorsed less than weekly smartphone use at night time. The authors mention that this could be because there were more older individuals in the follow-up sample. I wonder how effective these kinds of campaigns could be for younger individuals. This could be an interesting point to make for future directions.

• It could also be beneficial to include a note in the discussion about the potential long-term effects of public health campaigns such as this. Have past interventions given a sense for longevity of impact?

• Can the authors provide any estimates for the uptake of this intervention? It would be helpful to know how many people heard the public health messages, how many recalled information about smartphones and sleep conveyed in the public health campaign, etc.

Reviewer #2: This study reports interesting results from a mass media education campaign aiming to promote healthy sleep behaviors; the intervention included multiple modes through which information on nighttime smart phone use and sleep behavior was delivered to a large audience through radio programming and social media, with personalized feedback for survey respondents, and encouragement to participate in an online survey. Overall, the intervention is a nice example of an innovative mass media intervention to promote sleep hygiene. However, the manuscript currently has a few weaknesses that limit the contributions of this research to the larger evidence base, as described below.

Introduction

1. A large share of the framing of this study concerns the value of citizen science, yet it remains unclear whether this intervention is an example of citizen science. This terms is often used when a large number of the public is involved in the research process, generally data collection, categorizing, or other procedures. An adequate working definition is not provided for this term, as public participation in scientific research is too vague, and, more importantly, how this is an example of citizen science is needed. As I understand this intervention, I would categorize it as a mass media or public education campaign that is evaluated using a large survey.

2. The introduction references the increasing degree to which smartphones are integrated into society, but it need to provide additional evidence/citations for statements, such as “Smartphones are frequently used around the clock.”

Methods

3. Additional justification for the intervention is needed. For example, how was the personal feedback messaging chosen, and how might the role of age influence the effectiveness? The personal feedback appears to be using a social norms approach, but these are highly dependent on the age group of participants (which would arguably be expected to influence the effectiveness of this part of the intervention).

4. What was the specific messaging in the news articles, video, and interviews, and, in particular, how did the messaging/recommendations relate to the changes in nighttime smartphone use behavior that were assessed two weeks later? It is important to understand what behavioral changes would be expected if the intervention was a success.

a. An interesting future direction, if the messaging varied in important ways from day-to-day, would be to consider whether the specific behavioral changes varied based on the day participants completed their initial survey. This would assume that participants were not exposed to the education in other days, which may be too strong of an assumption. To be clear, I am not requesting these analyses be considered in this paper, rather I am just bringing up this possibility.

5. How might “The results of the day” intervention influence responses for participant who completed the survey (i.e., is there potential for priming or norms to bias reports)? However, this is likely a minor issue for this study due to the primary reliance on follow-up data, other than weekly night-time smartphone use.

6. Can you define news articles on page 126? I am wondering if these are just social media posts.

7. Typo on line 162, which should probably read as “More often”

8. Typo on line 178 for ‘classification.’ Also, can you define the ISCED categories?

9. Additional information is needed about baseline night-time smartphone use. For example, how was night-time defined for participants, and what is meant by weekly vs. less than weekly? So, a single time during the week counts as weekly?

10. Figure 1 is unhelpful. If you are to retain, then you need to restructure and provide additional information about the intervention components and potentially when they were delivered during the intervention week.

Results

11. The description of the education finding is that “More participants with high education did not change their smartphone habits compared to participants with low or medium education.” It looks like the finding is that more low or medium education participants tried to change habits but were unsuccessful relative to high education; this should be clarified.

12. It appears that baseline night-time use is by far the strongest predictor of who changes their habits, and so it would be useful to report tests of socio-demographic differences where you simultaneously adjust for these variables (including baseline night-time use). Maybe this explains the reason for the lower success by education categories.

13. The term ‘the sleep period’ is used in reference to the baseline data collection and the two-week follow-up, and so it is unclear what is meant.

14. The analysis plan is very simple, with simple calculations of percentages. This does not lend itself to drawing conclusions about differences between the various groups or the most common behavioral and motivational factors for changes.

Discussion

15. Limitations are mentioned, such as the use of self-report and non-representativeness of the sample. Given the use of a non-randomized trial and clear messaging around the desired/recommended smartphone behavior (increasing social desirability bias), and the lack of statistical tests, the authors need to be more cautious about drawing causal claims (i.e., had direct interventional effects).

16. On line 294, it is unclear what is meant by before and/or during the sleep period.

17. There is no reference to any other sleep-related literature in the Discussion. The authors need to do a more comprehensive literature search so that they can integrate their findings into the broader literature.

Reviewer #3: The current study investigated the SmartSleep Experiment targeting nighttime smart phone behaviors and use, a public health concern, in a sample of nearly 9000 Danish adolescents aged 16 years and above. The authors report that 9% of the study participants changed their nighttime smart phone habits as a result of the intervention; 78% of these indicated that they continue to do so at follow-up. Behavioral changes included activating silent mode, or reducing use before and during sleep.

The introduction is straightforward and focuses on implementing what the authors term public health citizen science projects to test an intervention called Smart sleep experiment among Danish adults. The authors highlight how smart phone use has become a public health concern as children as well as adults frequently use smart phones around the clock thus impacting the quality and duration of sleep. Thus the advocate for a citizen science project which addresses a public health concern on a large scale in a population of interest.

The methods are straightforward as well then describe data collection procedures, as well as human subjects protections considerations. The intervention consisted of participants receiving immediate feedback about nighttime smart phone use thus raising awareness among study participants with the goal or intention of reducing the same to promote higher quality and longer duration sleep. The study was accompanied by a public media campaign that included principally radio programs, the websites of the same as well as social media processes associated with radio stations based on the description provided. Following the initial assessment and feedback, a two-week follow-up survey was completed to ascertain the extent to which study participants made a change to their sleep hygiene.

The sample originally included over 25,000 Danish adults, half of which indicated that they would complete a follow up with an actual response rate of about 8900 participants. Data analyses focused principally on chi-square tests to examine changes in the distribution of variables as well as ANOVA was for continuous variables.

Study findings provided evidence of important differences between individuals who changed habits, try to change habits, or have not changed any sleep habits by sex, by educational level, as well as the extent to which smart phones were used at baseline, and by residential status, living alone versus not living alone. As noted, a bit over 800 participants are 9% of the sample change their nighttime smart phone habits following the intervention. The majority of participants made these changes because they wanted to improve on their sleep hygiene, reducing sleep problems and increasing positive sleep. Participants used different strategies to achieve changes that are detailed in the manuscript. The authors also tapped into understanding the underlying motivational factors as they labeled them, which among others included wanting to reduce sleep problems, wanting to reduce unhealthy smart phone habits, increase knowledge on health consequences of poor sleep, and discussion of smart phone habits with friends.

In conclusion, the current large-scale citizen science project, which partnered with a radio-based media campaign, that implemented raising awareness about how smart phone use interfered with proper sleep hygiene provided some promising preliminary evidence. The mechanism of change underlying the observed behavioral changes among study participants were largely related to raising awareness as well as novel insights as to how smart phone use might impact poor sleep as well as reduced sleep duration. Some inherent threats to the current study design includes the fact that over half of study participants were not willing to provide follow-up data and of those approximately 70% who agreed to do so provided data. There is no way of knowing the extent to which the current study findings are idiosyncratic and not representative simply because of this issue. This does not profoundly change the promise of the current effort but needs to be addressed adequately in the manuscript including perhaps some additional follow-up analyses of individuals who agreed to provide follow-up data initially versus ones who did not as well as between the group of individuals who agreed at baseline and then actually provided data versus the ones who did not. Doing so simply will instill greater confidence and study findings.

6. PLOS authors have the option to publish the peer review history of their article (what does this mean?). If published, this will include your full peer review and any attached files.

Reviewer #1: No

Reviewer #2: No

Reviewer #3: No

---

## [Author Response · Author response to Decision Letter 0]

15 Apr 2021

Editor’s comments

Please provide more details related to the results you obtained, while keeping the scientific rigor related to data presentation. Please discuss the limitations of the study and the impact of the research on the society.

Response: We have followed al of the recommendations from the editor and reviewers to the best of our abilities. Please see details below.

Response: We have adjusted the format including the file naming accordingly. 

Response: We have included additional details regarding participant consent in the method section, lines 101-102:

Written informed consent was obtained from all participants. The participants were informed about the purpose of the research and their rights to withdraw.

The data set contains personally identifiable and sensitive survey data information. We are therefore not allowed to make them publicity available according to the Danish Protection Agency (Danish data protection legislation (datatilsynet.dk)) and Danish law. Inquiries for secure data access under conditions stipulated by the Danish Data Protection Agency should be directed at the data manager of the SmartSleep project (liks@sund.ku.dk) or principle investigator of the SmartSleep project Professor Naja Hulvej Rod (nahuro@sund.ku.dk) 

4.We note that the grant information you provided in the ‘Funding Information’ and ‘Financial Disclosure’ sections do not match. When you resubmit, please ensure that you provide the correct grant numbers for the awards you received for your study in the ‘Funding Information’ section.

Response: We have updated the Funding Information and Financial Disclosure so the sections match.

The project was funded by the Independent Research Fund Denmark (grant number 7025-00005B). The funder had no role in study design, data collection and analysis, decision to publish, or preparation of the manuscript.

Reviewer 1

This manuscript examines the impact of a massive public media campaign on smartphone use at night in adults. This study suggests that public health efforts can be made through the mass media that can influence behavior, at least in the short term. I provide several questions regarding the study methods and conclusions along with recommendations for improving the paper below.

Response: Thank you for your constructive feedback. Please see our detailed response below.

Reviewer 1, Question 1

In the introduction, there is good discussion of the use of citizen science to conduct research, however this section could benefit from a discussion of the impact of smartphone use on sleep behavior and past interventions addressing this public health problem.

Response: Thank you for highlighting this. Only few previous interventions have directly addressed the impact of night-time smartphone use on sleep. However, updating our literature search we identified a recent randomized trial, which we have added to the discussion about the effects of night-time smartphone use on sleep, Introduction section lines 47-49:

Furthermore, a recent randomized trial with 38 college students found that restricting mobile phone use before bedtime reduced sleep latency and increased sleep duration [7].

Reviewer 1, Question 2

The study purports to assess intervention effects on both smartphone use at night and sleep behavior but none of the study measures assess sleep behavior (e.g., sleep duration, seep onset latency, etc.), making the second aim impossible.

Response: We agree, but we unfortunately have no information on sleep behavior in the follow-up questionnaire, and we can therefore not address whether the massive media campaign directly affected the participants’ sleep behavior. Thus, the aim of the paper is to assess whether the massive public campaign have an impact on the participants’ night-time smartphone use. We have tried to make this more clear in the introduction section:

Lines 51-53:

Thus, there is a need for potential public health prevention strategies to change night-time smartphone behavior to eventually improve sleep patterns and health.

Lines 73-74:

During one week of data collection, DR created a national mass media campaign focusing on night-time smartphone behaviors and sleep…

Lines 80-81:

2) examine behavioral and motivational factors that may have influenced their smartphone behavior changes…

Reviewer 1, Question 3

The method used for recruiting participants presents a significant limitation to drawing conclusions from the study findings. Because only those who indicated interest in participating in research following the citizen science intervention, a sample bias likely exits because people who agree to participate in the may have been more likely to have changed their behavior. It is difficult to conclude that the intervention produced the change given this sampling bias.

Response: We acknowledge that there is a risk of sampling bias in the follow-up study due to the recruitment strategy. We have discussed this limitation in the discussion section lines 372-375:

Moreover, participants who agreed to participate in the follow-up study may be more likely to change their smartphone behavior compared to those who did not want to participate in future studies, which may have resulted in a slight over estimation of the effect of the intervention.

Reviewer 1, Question 4

Another significant limitation is the inability to determine the size of the effect of the intervention on smartphone use at night. The four response choices in the measure do not allow for assessment of the size of the intervention effect.

Response: We agree that it is a limitation and a potential for future research. We have added a sentence to the discussion section lines 357-358:

… it would be interesting to explore the actual changes in night-time smartphone use including the size of the interventional effect on night-time smartphone use.

Reviewer 1, Question 5

The utility of the measure of smartphone use is unclear because there are no reported studies assessing the psychometrics of this measure. Relatedly, the paper could be improved by reporting reliability and validity for the current study sample.

Response: Measuring smartphone behavior is a major challenge as digital technology and smartphone use is rapidly changing, and the measures in previous studies quickly becomes outdated. Thus, we found it necessary to develop our own measure. We generally agree that it is beneficial to validate new measures, but given it is a single-item count measure and not a scale with multiple items, we find this less pressing/doable. In future work, it would be interesting to validate the self-reported night-time smartphone use with objective measures using sensor-driven smartphone tracking data. 

Reviewer 1, Question 6

Regarding the question assessing “behavioral interventions”, were participants allowed to select more than one of the options (e.g., used an analog clock as alarm AND placed my phone out of reach) or were they asked to select only one option? It could be helpful to clarify this with the “motivational factors” as well.

Response: Thank you for this comment. The participants were allowed to select more than one of the response options in both behavioral factors and motivational factors. We have clarified this in the material section line 183 and line 189. 

The participants could select multiple response options. 

Reviewer 1, Question 7

Is there any additional information the authors could give about the rationale for including age, gender, and education level in the survey while they do not report other possible variables that could impact sleep (e.g., sleep disorders; socioeconomic status)?

Response: Thank you for these relevant suggestions. We have added education and occupational status (as measures of socio-economic status) and sleep quality to Table 1, as suggested. We unfortunately did not have information on diagnosed sleep disorders. 

Please see the updated Table 1 at the end of the rebuttal.

Reviewer 1, Question 8

It is interesting how many people in the study sample endorsed less than weekly smartphone use at night time. The authors mention that this could be because there were more older individuals in the follow-up sample. I wonder how effective these kinds of campaigns could be for younger individuals. This could be an interesting point to make for future directions.

Response: We agree that it would be interesting to investigate whether a mass media campaign like the SmartSleep Experiment will be more effective in terms of changing health behavior among younger individuals. Please see the Discussion section line 377-379:

For future studies, it would be interesting to explore whether using mass media campaigns to improve behavior changes would be more/less effective in younger populations.

Reviewer 1, Question 9

It could also be beneficial to include a note in the discussion about the potential long-term effects of public health campaigns such as this. Have past interventions given a sense for longevity of impact?

Response: It is unfortunately difficult to estimate the long-term effects of the campaign, but a review investigating the health behavior effects from mass media campaign proposes that longer and more intensive campaigns may be more effective in changing long-term health behavior. We have included a note about this in the Discussion lines 348-351: 

Furthermore, while short-term behavior changes may be achieved using mass media campaigns and citizen science, the long-term effects may be more difficult to maintain. It has previously been proposed that longer and more intense mass media campaign are likely to be more effective to maintain long-term behavior changes [19].

Reviewer 1, Question 10

Can the authors provide any estimates for the uptake of this intervention? It would be helpful to know how many people heard the public health messages, how many recalled information about smartphones and sleep conveyed in the public health campaign, etc.

Response: We have information on how many unique listeners listened to the radio channels (Good morning P3 and P4) during the week of the SmartSleep Experiment, which is in the material section. As there were a lot of interviews, teasers, discussions etc. about smartphone use and sleep during the mass media campaign on these radio channels, this may be a good estimate of how many people have heard about the SmartSleep Experiment. However, we do not have information on how many listeners actually recalled the information from the public health campaign. We added this limitation in our Discussion line 366-370:

All participants in the SmartSleep Experiment were exposed to at least parts of the media campaign, as this was the platform from which they were recruited. Unfortunately, we do not have specific information on how many elements of the campaign each person were exposed to, or how much of the information they recalled after the two week follow-up period.

Reviewer #2: 

This study reports interesting results from a mass media education campaign aiming to promote healthy sleep behaviors; the intervention included multiple modes through which information on nighttime smart phone use and sleep behavior was delivered to a large audience through radio programming and social media, with personalized feedback for survey respondents, and encouragement to participate in an online survey. Overall, the intervention is a nice example of an innovative mass media intervention to promote sleep hygiene. However, the manuscript currently has a few weaknesses that limit the contributions of this research to the larger evidence base, as described below.

Response: Thank you for your constructive feedback and criticism. We have tried to address each of your specific comments below.

Reviewer 2, Question 1

A large share of the framing of this study concerns the value of citizen science, yet it remains unclear whether this intervention is an example of citizen science. This terms is often used when a large number of the public is involved in the research process, generally data collection, categorizing, or other procedures. An adequate working definition is not provided for this term, as public participation in scientific research is too vague, and, more importantly, how this is an example of citizen science is needed. As I understand this intervention, I would categorize it as a mass media or public education campaign that is evaluated using a large survey.

Response: Thank you for raising this discussion. The participants in this study were directly involved in the data collection, wherefore this study is an example of a citizen science study and not only a mass media campaign. There are several ways to involve the public in the research process and the extent of involvement may differ in citizen science project [10]. We acknowledge that participants in this study may not be highly involved in the project; however, we will still categorize it as citizen science as participants were an active part of the data collection. 

We agree that public participation in scientific research may be too vague for a definition of citizen science and we have added a clearer definition of citizen science in our Introduction lines 54-55:

Citizen science, defined as public participation in scientific research in which the public are engaged directly in one or more of the research processes [10,11]…

Furthermore, we have added a sentence on how the participants were actively involved in the data collection in the method section lines 88-91:

The participants were actively involved in the data collection by filling out a survey, which was an integrated part of the media campaign. This was combined with direct individual feedback to the participants and real-time presentation of preliminary results from the data collection to all radio listeners during the week of the campaign. 

Reviewer 2, Question 2

The introduction references the increasing degree to which smartphones are integrated into society, but it need to provide additional evidence/citations for statements, such as “Smartphones are frequently used around the clock.”

Response: We have added additional references to support the statement. 

Reviewer 2, Question 3

Additional justification for the intervention is needed. For example, how was the personal feedback messaging chosen, and how might the role of age influence the effectiveness? The personal feedback appears to be using a social norms approach, but these are highly dependent on the age group of participants (which would arguably be expected to influence the effectiveness of this part of the intervention).

Response: We acknowledge that there may be several ways of giving personal feedback, which may influence the effectiveness of the intervention. For this specific project, the personal feedback was merely considered a feature to motivate people to participate in the survey, and it only played a smaller role in the overall media campaign. While we acknowledge that there is a whole literature on different approaches to personal feedback interventions, focusing on this element was not the purpose of the current study. We have clarified the minor motivational role of the personal feedback in the current study to prevent confusion, in the method section, lines 105-106:

To motivate people to participate in the survey, they received immediate personal feedback on their night-time smartphone use after completing the survey. 

Reviewer 2, Question 4

What was the specific messaging in the news articles, video, and interviews, and, in particular, how did the messaging/recommendations relate to the changes in nighttime smartphone use behavior that were assessed two weeks later? It is important to understand what behavioral changes would be expected if the intervention was a success.

Response: The messaging in the news and feature articles, video, and interviews were generally focused on the importance of sleep for health and well-being and on how night-time smartphone use could lead to disturbed sleep. We were unfortunately not able to identify the individual effects of each of these elements, and the presented results should be seen as evaluation of the full media campaign. We have highlighted this in the method section lines 136-139:

More than 20 news and feature articles focusing on the importance of sleep for health and well-being and on how night-time smartphone use may result in disturbed sleep were published during the week of the experiment on DR's website and their social media pages, including Twitter, Facebook, and Instagram.

Reviewer 2, Question 5

An interesting future direction, if the messaging varied in important ways from day-to-day, would be to consider whether the specific behavioral changes varied based on the day participants completed their initial survey. This would assume that participants were not exposed to the education in other days, which may be too strong of an assumption. To be clear, I am not requesting these analyses be considered in this paper, rather I am just bringing up this possibility.

Response: This is an interesting suggestion, but the theme and the interviews did not vary from day to day. Also, the majority of the listeners would probably tune in on multiple days as this a very popular morning show, and it would be difficult to tease out the individual effects of the elements presented one day as opposed to another. 

Reviewer 2, Question 6

How might “The results of the day” intervention influence responses for participant who completed the survey (i.e., is there potential for priming or norms to bias reports)? However, this is likely a minor issue for this study due to the primary reliance on follow-up data, other than weekly night-time smartphone use.

Response: ‘The results of the day’ were actually used to improve representation of the study population e.g. gender and age distributions and geographical representation. This means that we encourage people who were less represented in the survey to participate e.g. male, specific age groups, specific educational levels etc. 

Reviewer 2, Question 7

Can you define news articles on page 126? I am wondering if these are just social media posts.

The articles referring to were news articles and feature articles at DR’s webpage (www.dr.dk) and not just social media posts. The articles included e.g. results from previous research on sleep disorders or how the blue light from smartphones may affect sleep. We have clarified this in the method section lines 136-139: 

More than 20 news and feature articles focusing on how night-time disturbances, including night-time smartphone use, could lead to disturbed sleep and the importance of sleep for your health were published during the week of the experiment on DR’s website and their social media pages, including Twitter, Facebook, and Instagram. 

Reviewer 2, Question 8

Typo on line 162, which should probably read as “More often”

Response: Thank you for noticing. It has been corrected. 

Reviewer 2, Question 9

Typo on line 178 for ‘classification.’ Also, can you define the ISCED categories?

Response: Thank you for noticing the typo. We have added more information on the categorization of educational level including a reference to the ISCED 2011 in the method section lines 190-192:

educational level (low (primary school); medium (upper secondary school; technical vocational education); high education (short, medium, and long cycle higher education); and other based on the International Standard Classification of Education 2011 [25]

Reviewer 2, Question 10

Additional information is needed about baseline night-time smartphone use. For example, how was night-time defined for participants, and what is meant by weekly vs. less than weekly? So, a single time during the week counts as weekly?

Response: We define night-time smartphone use as any short-term and/or long-term activation of the smartphone after falling asleep within the past three month with four response options ranging from every night or almost every night to never. We have added additional information about the definition of night-time smartphone use in the manuscript. For descriptive purposes, we have categorized baseline night-time smartphone use into two categories: weekly (every night or almost every night and a few nights a week) and less than weekly (a few nights a month or less and never). 

Method section lines 195-199:

Baseline night-time smartphone use was assessed by asking how often the smartphone was used after falling asleep within the past three months with the following response options: every night or almost every night; a few nights a week;, a few nights a month or less; and never. Smartphone use was defined in the baseline survey and referred to both short and long activation of the smartphone.

Reviewer 2, Question 11

Figure 1 is unhelpful. If you are to retain, then you need to restructure and provide additional information about the intervention components and potentially when they were delivered during the intervention week.

Response: We have merged the information from Figure 1 and 2 into the new figure 

Please see the updated figure at the end of the rebuttal. 

Reviewer 2, Question 12

The description of the education finding is that “More participants with high education did not change their smartphone habits compared to participants with low or medium education.” It looks like the finding is that more low or medium education participants tried to change habits but were unsuccessful relative to high education; this should be clarified.

Response: We agree and we have added a sentence for clarification in the result section lines 245-246:

Additionally, more participants with low or medium education tried to change their smartphone habits, but were unsuccessful compared to participants with high education. 

Reviewer 2, Question 13

It appears that baseline night-time use is by far the strongest predictor of who changes their habits, and so it would be useful to report tests of socio-demographic differences where you simultaneously adjust for these variables (including baseline night-time use). Maybe this explains the reason for the lower success by education categories.

Response: We have compared night-time smartphone use across groups with different educational level and occupational status in Supporting Material table 2. We have briefly commented on these differences in the revised manuscript lines 252-255:

S2 Table shows the socio-demographic differences in baseline night-time smartphone use. It appears that participants with high education and employed people are less likely to use their smartphones during sleep hours several times a week or more, while students are more likely to use their smartphones during sleep hours.

Reviewer 2, Question 14

The term ‘the sleep period’ is used in reference to the baseline data collection and the two-week follow-up, and so it is unclear what is meant.

Response: The sleep period refers to sleep hours when the participants would typically sleep. We have modified the description of the term throughout the manuscript.

Reviewer 2, Question 15

The analysis plan is very simple, with simple calculations of percentages. This does not lend itself to drawing conclusions about differences between the various groups or the most common behavioral and motivational factors for changes

Response: While we appreciate the value of descriptive analysis in a relatively unexplored field, we agree and have modified our conclusion to be less conclusive.

Abstract lines 37-38:

Using citizen science and mass media appeared to have some interventional impacts on night-time smartphone behavior.

Discussion section lines 296-298:

Furthermore, we find that this mass media campaign and the citizen science approach with direct interaction between scientists and radio listeners during the SmartSleep Experiment appeared to have some interventional effects on the participants' night-time smartphone behavior.

Line 383-386: This study shows that massive media attention and the citizen science approach with direct interaction between scientists and radio listeners may have had some interventional effects on night-time smartphone behaviors. Increased knowledge and awareness seemed to be the key motivational drivers.

Reviewer 2, Question 16

Limitations are mentioned, such as the use of self-report and non-representativeness of the sample. Given the use of a non-randomized trial and clear messaging around the desired/recommended smartphone behavior (increasing social desirability bias), and the lack of statistical tests, the authors need to be more cautious about drawing causal claims (i.e., had direct interventional effects).

Response: We agree, and we have modified our conclusion to be less conclusive. 

Abstract lines 37-38:

Using citizen science and mass media may appear to have some interventional impacts on night-time smartphone behavior.

Discussion lines 296-298:

Furthermore, we find that this mass media campaign and the citizen science approach with direct interaction between scientists and radio listeners during the SmartSleep Experiment may appear to have some interventional effects on the participants' night-time smartphone behavior.

Line 383-386: This study shows that massive media attention and the citizen science approach with direct interaction between scientists and radio listeners may have had some interventional effects on night-time smartphone behaviors. Increased knowledge and awareness seemed to be the key motivational drivers.

Reviewer 2, Question 17

On line 294, it is unclear what is meant by before and/or during the sleep period.

Response: We have added more information to clarify. 

Lines 327-330: Around half of the participants who changed night-time smartphone behavior stated that they have reduced their smartphone use before falling asleep at night and during sleep hours when they would typically sleep.

Reviewer 2, Question 18

There is no reference to any other sleep-related literature in the Discussion. The authors need to do a more comprehensive literature search so that they can integrate their findings into the broader literature.

Response: We have included additional sleep-related literature in the discussion section lines 315-322:

Sleep problems are a prevalent and increasing public health issue in adult populations [27, 28] and evidence has shown that sleep problems are related to adverse short- and long-term health effects including poor mental health, risk-taking behavior, cardiovascular diseases, diabetes, and mortality [8, 9]. Furthermore, several studies have shown that night-time smartphone use is related to poor sleep [1, 5, 6]. The massive public focus on sleep and night-time smartphone use in the mass media during the SmartSleep Experiment may have increased awareness of sleep and smartphone habits, which may be an effective strategy to improve sleep behavior [29].

Reviewer #3: 

The current study investigated the SmartSleep Experiment targeting nighttime smart phone behaviors and use, a public health concern, in a sample of nearly 9000 Danish adolescents aged 16 years and above. The authors report that 9% of the study participants changed their nighttime smart phone habits as a result of the intervention; 78% of these indicated that they continue to do so at follow-up. Behavioral changes included activating silent mode, or reducing use before and during sleep.

The introduction is straightforward and focuses on implementing what the authors term public health citizen science projects to test an intervention called Smart sleep experiment among Danish adults. The authors highlight how smart phone use has become a public health concern as children as well as adults frequently use smart phones around the clock thus impacting the quality and duration of sleep. Thus the advocate for a citizen science project which addresses a public health concern on a large scale in a population of interest.

The methods are straightforward as well then describe data collection procedures, as well as human subjects protections considerations. The intervention consisted of participants receiving immediate feedback about nighttime smart phone use thus raising awareness among study participants with the goal or intention of reducing the same to promote higher quality and longer duration sleep. The study was accompanied by a public media campaign that included principally radio programs, the websites of the same as well as social media processes associated with radio stations based on the description provided. Following the initial assessment and feedback, a two-week follow-up survey was completed to ascertain the extent to which study participants made a change to their sleep hygiene.

The sample originally included over 25,000 Danish adults, half of which indicated that they would complete a follow up with an actual response rate of about 8900 participants. Data analyses focused principally on chi-square tests to examine changes in the distribution of variables as well as ANOVA was for continuous variables.

Study findings provided evidence of important differences between individuals who changed habits, try to change habits, or have not changed any sleep habits by sex, by educational level, as well as the extent to which smart phones were used at baseline, and by residential status, living alone versus not living alone. As noted, a bit over 800 participants are 9% of the sample change their nighttime smart phone habits following the intervention. The majority of participants made these changes because they wanted to improve on their sleep hygiene, reducing sleep problems and increasing positive sleep. Participants used different strategies to achieve changes that are detailed in the manuscript. The authors also tapped into understanding the underlying motivational factors as they labeled them, which among others included wanting to reduce sleep problems, wanting to reduce unhealthy smart phone habits, increase knowledge on health consequences of poor sleep, and discussion of smart phone habits with friends.

In conclusion, the current large-scale citizen science project, which partnered with a radio-based media campaign, that implemented raising awareness about how smart phone use interfered with proper sleep hygiene provided some promising preliminary evidence. The mechanism of change underlying the observed behavioral changes among study participants were largely related to raising awareness as well as novel insights as to how smart phone use might impact poor sleep as well as reduced sleep duration. Some inherent threats to the current study design includes the fact that over half of study participants were not willing to provide follow-up data and of those approximately 70% who agreed to do so provided data. There is no way of knowing the extent to which the current study findings are idiosyncratic and not representative simply because of this issue. This does not profoundly change the promise of the current effort, but needs to be addressed adequately in the manuscript including perhaps some additional follow-up analyses of individuals who agreed to provide follow-up data initially versus ones who did not as well as between the group of individuals who agreed at baseline and then actually provided data versus the ones who did not. Doing so simply will instill greater confidence and study findings

Response: Thank you for your helpful comments and feedback. We have added some additional analyses to investigate the representativeness of the study population in Supplemental information Table 1 in the method section lines 160-165:

S1 Table shows characteristics of individuals who at baseline did not agree to participate in future studies (n=12,787), individuals who agreed at baseline but did not participate in the follow-up study (n=3,437), and individuals who participated in the follow-up study (n=8,911). Those who participated in the follow-up study were generally older, had higher educational level, and were slightly less likely to use their smartphone during sleep hours than those who did not participate. 

References:

[1] Rod, N.H., et al., Overnight smartphone use: A new public health challenge? A novel study design based on high-resolution smartphone data. PLoS One, 2018. 13(10): p. e0204811

[5] Christensen, M.A., et al., Direct Measurements of Smartphone Screen-Time: Relationships with Demographics and Sleep. PLoS One, 2016. 11(11).

[6] Thomee, S., Mobile Phone Use and Mental Health. A Review of the Research That Takes a Psychological Perspective on Exposure. Int J Environ Res Public Health, 2018. 15(12): p. 2692.

[7] He, J.-W., et al., Effect of restricting bedtime mobile phone use on sleep, arousal, mood, and working memory: A randomized pilot trial. PloS one, 2020. 15(2): p. e0228756-e0228756.

[8] Medic, G., M. Wille, and M.E. Hemels, Short- and long-term health consequences of sleep disruption. Nat Sci Sleep, 2017. 9: p. 151-161.

 [9] Itani, O., et al., Short sleep duration and health outcomes: a systematic review, meta-analysis, and meta-regression. Sleep Med, 2017. 32: p. 246-256.

[10] Shirk, J., et al., Public Participation in Scientific Research: a Framework for Deliberate Design. Ecology and Society, 2012. 17(2).

[11] Bonney, R., et al., Citizen Science: A Developing Tool for Expanding Science Knowledge and Scientific Literacy. Bioscience, 2009. 59(11): p. 977-984.

[19] Wakefield, M.A., B. Loken, and R.C. Hornik, Use of mass media campaigns to change health behaviour. Lancet (London, England), 2010. 376(9748): p. 1261-1271.

[25] Schneider, S., The International Standard Classification of Education 2011. Comparative Social Research, 2013. 30: p. 365-379.

[27] Ferrie, J.E., et al., Sleep epidemiology--a rapidly growing field. Int J Epidemiol, 2011. 40(6): p. 1431-7.

[28] Chattu, V.K., et al., The Global Problem of Insufficient Sleep and Its Serious Public Health Implications. Healthcare (Basel, Switzerland), 2018. 7(1).

[29] Filip, I., et al., Public health burden of sleep disorders: underreported problem. Journal of Public Health, 2017. 25(3): p. 243-248.

---

## [Decision Letter · Decision Letter 1]

24 May 2021

PONE-D-21-02819R1

The SmartSleep Experiment: The interventional effects of using a citizen science approach and mass media to change smartphone and sleep behaviors

PLOS ONE

Dear Dr. Andersen,

Thank you for submitting your manuscript to PLOS ONE. After careful consideration, we feel that it has merit but does not fully meet PLOS ONE’s publication criteria as it currently stands. Therefore, we invite you to submit a revised version of the manuscript that addresses the points raised during the review process.

In the revised version, please consider the comments made by Reviewer #2 listed below.

We look forward to receiving your revised manuscript.

Kind regards,

Camelia Delcea

Academic Editor

PLOS ONE

Journal Requirements:

Reviewers' comments:

Reviewer's Responses to Questions

**Comments to the Author**

1. If the authors have adequately addressed your comments raised in a previous round of review and you feel that this manuscript is now acceptable for publication, you may indicate that here to bypass the “Comments to the Author” section, enter your conflict of interest statement in the “Confidential to Editor” section, and submit your "Accept" recommendation.

Reviewer #1: All comments have been addressed

Reviewer #2: (No Response)

Reviewer #3: All comments have been addressed

2. Is the manuscript technically sound, and do the data support the conclusions?

Reviewer #1: Yes

Reviewer #2: Partly

Reviewer #3: Yes

3. Has the statistical analysis been performed appropriately and rigorously? 

Reviewer #1: Yes

Reviewer #2: No

Reviewer #3: Yes

4. Have the authors made all data underlying the findings in their manuscript fully available?

Reviewer #1: No

Reviewer #2: (No Response)

Reviewer #3: Yes

5. Is the manuscript presented in an intelligible fashion and written in standard English?

Reviewer #1: Yes

Reviewer #2: (No Response)

Reviewer #3: Yes

6. Review Comments to the Author

Reviewer #1: (No Response)

Reviewer #2: I appreciate the revisions the authors made to improve the clarity of the methods and address study limitations.

For the most part, the authors adequately responded to my comments. The few exceptions are as follows:

1. in my opinion, the most substantial limitation is that there was no risk-adjustment made based on baseline nightly smartphone use when reporting results of the intervention and demographic differences. For instance, 40% of participants who regularly use their smartphone at night changed or attempted to change habits relative to 11% of respondents who used their smartphone at night several nights a month or less. These data suggest baseline nightly smartphone use is a large predictor of whether or not people attempt change. The second paragraph of the results now mentions differences in the outcome as well as sociodemographic characteristics by baseline night-time smartphone use; this is very helpful. However, it remains unknown the extent to which differences in who is changing or attempting to change nightly smartphone habits are merely an artifact of baseline use. This is a problem particularly if the interpretation suggests differences in the effectiveness of the intervention by demographic differences, but these likely partly if not completely stem from baseline risks (e.g., do younger folks attempt/successfully change habits only because they are much more likely to initially have problems with nightly use?). Another example where this comes up is how the data are generally interpreted (e.g., "still 85% of the study population indicated that they had not changed night-time smartphone behavior), which suggests that lack of change is a failure of the intervention. Some might not have anywhere to change to in that they reported no use before. It is thus essential that some risk-adjustment is made and that the data are interpreted based on who has potential to change due to baseline problems.

2. causal language is still used throughout including in the title ("interventional effects") despite this study being descriptive and susceptible to bias from sampling and self-reported measures. The descriptive nature of this study should be explicit.

3. there is no recognition of treatment heterogeneity based on age. Whether or not the personal feedback was included strictly as a motivational tool, the feedback was highly age-dependent and specific to use of smartphone at night and therefore may have influenced age-specific findings in likelihood of changing nighttime smartphone use. This should at the very least be briefly mentioned as a limitation or future direction when interpreting age-specific findings.

Also, the language is generally clear. But, there are some grammatical errors throughout. The results could also be edited for clarity, particularly the first two paragraphs.

Reviewer #3: The authors have been fairly responsive to the reviewer feedback, no additional comments at this time

7. PLOS authors have the option to publish the peer review history of their article (what does this mean?). If published, this will include your full peer review and any attached files.

Reviewer #1: No

Reviewer #2: No

Reviewer #3: **Yes: **Alexander T. Vazsonyi

---

## [Author Response · Author response to Decision Letter 1]

7 Jun 2021

Editor’s comment:

Response: We have made some corrections to the reference list, which should now be complete and correct. 

Reviewer #2

I appreciate the revisions the authors made to improve the clarity of the methods and address study limitations. For the most part, the authors adequately responded to my comments. The few exceptions are as follows:

1. In my opinion, the most substantial limitation is that there was no risk-adjustment made based on baseline nightly smartphone use when reporting results of the intervention and demographic differences. For instance, 40% of participants who regularly use their smartphone at night changed or attempted to change habits relative to 11% of respondents who used their smartphone at night several nights a month or less. These data suggest baseline nightly smartphone use is a large predictor of whether or not people attempt change. The second paragraph of the results now mentions differences in the outcome as well as sociodemographic characteristics by baseline night-time smartphone use; this is very helpful. However, it remains unknown the extent to which differences in who is changing or attempting to change nightly smartphone habits are merely an artifact of baseline use. This is a problem particularly if the interpretation suggests differences in the effectiveness of the intervention by demographic differences, but these likely partly if not completely stem from baseline risks (e.g., do younger folks attempt/successfully change habits only because they are much more likely to initially have problems with nightly use?). Another example where this comes up is how the data are generally interpreted (e.g., "still 85% of the study population indicated that they had not changed night-time smartphone behavior), which suggests that lack of change is a failure of the intervention. Some might not have anywhere to change to in that they reported no use before. It is thus essential that some risk-adjustment is made and that the data are interpreted based on who has potential to change due to baseline problems.

Response: This is a valid point. In order to take the baseline night-time smartphone use into account in our interpretation of the results, we have restricted all analyses to the participants who report using their smartphone during sleep at baseline (n=4,926 participants). We have updated Figure 2-4 and corrected the results throughout the paper. The overall findings are unchanged. 

2. Causal language is still used throughout including in the title ("interventional effects") despite this study being descriptive and susceptible to bias from sampling and self-reported measures. The descriptive nature of this study should be explicit.

Response: We changed the title accordingly: The SmartSleep Experiment: Evaluation of changes in night-time smartphone behavior following a mass media citizen science intervention.

Furthermore, we have included a sentence about the lack of causal inference in the discussion section lines 358-362: 

Even though we cannot determine the direct interventional effects from participating in the SmartSleep Experiment, the present findings give important insights into the connection between citizen science, mass media and interventional behavior impacts, which may be of value in future prevention strategies to improve sleep and smartphone habits.

3. There is no recognition of treatment heterogeneity based on age. Whether or not the personal feedback was included strictly as a motivational tool, the feedback was highly age-dependent and specific to use of smartphone at night and therefore may have influenced age-specific findings in likelihood of changing nighttime smartphone use. This should at the very least be briefly mentioned as a limitation or future direction when interpreting age-specific findings.

Response: We agree, and we have mentioned this limitation in the discussion section lines 375-377:

Furthermore, the direct personal feedback after completing the survey varied across age groups, and this may have influenced age-specific findings in likelihood of changing night-time smartphone use.

4. Also, the language is generally clear. But, there are some grammatical errors throughout. The results could also be edited for clarity, particularly the first two paragraphs.

Response: Thank you for highlighting this. We have edited the result section and corrected the grammatical errors throughout the manuscript. The manuscript has also been proof read by a professional proof reader.

---

## [Editor Report · Decision Letter 2]

14 Jun 2021

The SmartSleep Experiment: Evaluation of changes in night-time smartphone behavior following a mass media citizen science campaign

PONE-D-21-02819R2

Dear Dr. Andersen,

We’re pleased to inform you that your manuscript has been judged scientifically suitable for publication and will be formally accepted for publication once it meets all outstanding technical requirements.

Kind regards,

Camelia Delcea

Academic Editor

PLOS ONE
---

## [Editor Report · Acceptance letter]

24 Jun 2021

PONE-D-21-02819R2 

*The SmartSleep Experiment*: Evaluation of changes in night-time smartphone behavior following a mass media citizen science campaign 

Dear Dr. Andersen:

I'm pleased to inform you that your manuscript has been deemed suitable for publication in PLOS ONE. Congratulations! Your manuscript is now with our production department. 

Kind regards, 

on behalf of

Dr. Camelia Delcea 

Academic Editor

PLOS ONE